# Congruent-Arc Latarjet Using Subscapularis Split Approach in the Treatment of Anterior Shoulder Instability with Significant Bone Loss: A Case Series

**DOI:** 10.3390/healthcare13141768

**Published:** 2025-07-21

**Authors:** Ahmed Farid Mekky, Chiara Fossati, Alessandra Menon, Paolo Fici, Pietro Simone Randelli, Tarek Aly

**Affiliations:** 1Knee and Shoulder Arthroscopy Unit, Department of Orthopedic Surgery, Tanta University, El-Geish Street, Tanta 31527, Egypt; ahmed.mekky2020@yahoo.it; 2U.O.C. 1º Clinica Ortopedica, ASST Gaetano Pini-CTO, Piazza Cardinal Ferrari 1, 20122 Milan, Italy; chiara.fossati@hotmail.com (C.F.); ale.menon@me.com (A.M.); pietro.randelli@unimi.it (P.S.R.); 3Laboratory of Applied Biomechanics, Department of Biomedical Sciences for Health, Università degli Studi di Milano, Via Mangiagalli 31, 20133 Milan, Italy; 4Scuola di Specializzazione in Statistica Sanitaria e Biometria, Dipartimento di Scienze Cliniche e di Comunità, Università degli Studi di Milano, 20122 Milan, Italy; 5Scuola di Specializzazione in Ortopedia e Traumatologia, Università degli Studi di Milano, 20122 Milan, Italy; 6Research Center for Adult and Pediatric Rheumatic Diseases (RECAP-RD), Department of Biomedical Sciences for Health, Università degli Studi di Milano, Via Mangiagalli 31, 20133 Milan, Italy; 7Department of Orthopedic Surgery, Tanta University School of Medicine, Tanta 31527, Egypt; tarek.ali@med.tanta.edu.eg

**Keywords:** shoulder joint/surgery, joint instability, osteoarthritis, shoulder dislocation, bone loss, shoulder injuries, latarjet procedure, hill-sachs lesion

## Abstract

**Background:** Recurrent anterior shoulder instability is a common problem and may be associated with glenoid bone defects. Surgical procedures, including Latarjet, are the usual treatment for anterior shoulder instability, associated with significant glenoid bone defects. The aim of this study was to evaluate the clinical outcome and glenohumeral arthritis progression in patients with recurrent anterior shoulder instability and significant bone loss treated by a modified Latarjet procedure. **Methods:** From July 2018 to November 2021, a prospective observational case series was carried out on 21 patients with recurrent anterior shoulder instability associated with significant bone defects treated by a modified Latarjet procedure in which the coracoid process was rotated 90° on its longitudinal axis and the subscapularis muscle was horizontally split. Patients with a glenoid defect of more than 21% were included. Post-operatively, the patients were clinically assessed using modified Rowe scoring. Glenohumeral arthritis, graft position, union, and resorption were radiologically evaluated. **Results:** The mean age at the time of surgery was 28.52 ± 8.0 (range: 19–45) years. The mean number of dislocations was 18.33 ± 8.67 (range: 6–35) times. The mean glenoid defect size was 26.19 ± 4.85 (range: 21–37) % and Hill–Sachs lesions were off-track in 19 cases. The mean follow-up period was 30.67 ± 7.53 (range: 16–40) months. Eighteen patients (85.7%) showed good to excellent results. The mean modified Rowe score was 85.00 ± 18.77 (range: 30–100) points. The mean external rotation loss was 8.09 ± 5.11° (range: 0–20°). No cases of recurrent instability were observed, and there was no progression of glenohumeral arthritis. **Conclusions:** The modified Latarjet is an effective and reliable surgical option to treat traumatic anterior shoulder instability with significant bone loss. Most of the reported complications associated with this procedure did not affect the functional outcome.

## 1. Introduction

Post-traumatic anterior glenohumeral (GH) instability is a common problem which affects, in particular, young and sportive males with an incidence of 23.92 per 100,000 persons/year [1]. Surgical options for shoulder instability include anatomical interventions such as Bankart repair and non-anatomical ones such as the Latarjet procedure [2]. Arthroscopic capsulolabral repair is considered the most common procedure used to treat recurrent GH instability with good results in most patients but some factors such as young age, competitive and contact athletes, shoulder hyperlaxity, a large Hill–Sachs (HS) lesion and glenoid defect may result in a high failure rate and therefore bone procedure as Latarjet is an effective substitute. [3]. The lower bone defect threshold for which isolated capsulolabral repair is indicated remains controversial. Burkhart SS, De Beer JF [4], and Bigliani et al. [5] found that the acceptable glenoid defect for which the arthroscopic repair would be possible was 20% and 25%, respectively, and according to Itoi et al. [6], it was 21%. Balg and Boileau proposed that an instability severity index score (ISIS) ˃ 6 was recommended to undergo the bone procedure [7]. The Latarjet procedure involves osteotomy of the coracoid process and its transfer to the anterior border of the glenoid to address both bone and soft tissue deficiencies with a triple-blocking effect [8]: the extension of the glenoid arc by coracoid bone block, the sling effect of the conjoined tendon on abduction and external rotation (ABER), and capsular reinforcement by coracoacromial ligament (CAL) [9].

In the traditional Latarjet technique, originally described by Latarjet [10], the coracoid process is osteotomized at its axilla, and its undersurface lies in contact with the anterior glenoid through subscapularis tenotomy. Although the traditional procedure showed excellent results in terms of shoulder stability [11,12], the use of this technique is still controversial due to a possible loss of external rotation and an increased incidence of post-operative GH arthritis, which seems to be explained by lack of a close match between the glenoid and the coracoid lateral edge [13].

To exceed the limits of standard Latarjet, De Beer [14] modified the graft fixation technique to “congruent arc Latarjet” where the coracoid is rotated 90° on its longitudinal axis so its inferior surface lies in line with the articular surface of the glenoid. The subscapularis muscle is horizontally split for early recovery and rehabilitation, including external rotation [8]. It also allows treating significantly greater glenoid defects because the coracoid is wider than it is thick [15] (Figure 1). 

There are currently few studies reporting the clinical outcomes of the congruent-arc Latarjet. Therefore, the aim of this study was to evaluate clinical results and GH arthritis progression in patients with chronic anterior shoulder instability and significant bone loss, treated by a modified Latarjet procedure.

## 2. Methods

This prospective observational case series was approved by the Ethical Research Committee of the School of Medicine of Tanta University. All patients provided written informed consent prior to participation. Between July 2018 and November 2021, 27 consecutive patients with chronic recurrent anterior shoulder instability and significant glenoid bone loss (>21%) were screened.

### 2.1. Inclusion and Exclusion Criteria

Patients aged over 18 years with anterior shoulder instability persisting for more than 3 months and associated with critical glenoid bone loss (>21%), with or without an off-track Hill–Sachs lesion, were included.

Exclusion criteria were as follows: first-time dislocation, glenoid bone loss <21%, isolated Bankart lesion, posterior/multidirectional/voluntary instability, advanced glenohumeral osteoarthritis, rotator cuff tear, deltoid palsy, and generalized ligamentous hyperlaxity (Table 1).

### 2.2. Pre-Operative Evaluation

All patients were positive in apprehension and relocation tests. The Sulcus test and Beighton score were assessed to exclude multidirectional instability and generalized hyperlaxity. The modified Rowe score [16] and shoulder range of motion were evaluated. A standard anteroposterior (A/P) X-ray was used to detect pre-operative GH arthritis according to Samilson and Prieto classification [17]. Moreover, a computed tomography (CT) 3D scan was performed to determine the glenoid defect size through the best fit circle width loss method, the Hill–Sachs (HS) engagement through the on/off-track method (glenoid track concept), and the coracoid width and thickness. Lastly, patients underwent an MRI to assess potential labral, rotator cuff, and capsular injuries (Table 2).

### 2.3. Surgical Technique

The patients were placed in a beach chair position and a standard deltopectoral approach was used. The coracoid process was exposed, the CAL was cut 1 cm from its lateral aspect, and the pectoralis minor tendon was released from its medial aspect. The coracoid osteotomy was performed at the junction between the horizontal and vertical parts (the “knee” of the coracoid) by a 90° oscillating saw. The coracoid graft was prepared by removing a thin layer of bone from its medial surface, rotated 90° around its long axis, and grasped using the coracoid drill guide, which allows drilling 2 parallel 2.5 mm holes centered on the graft and perpendicular to the prepared medial surface about 1 cm apart.

With the arm in ABER, the subscapularis was horizontally split at the junction of its superior third and inferior two-thirds and then retracted by a Gelpi retractor. A Hohmann retractor was placed inferiorly to protect the axillary nerve, and a lever retractor was placed medial to the glenoid rim. The capsule was vertically incised and marked with sutures to facilitate its identification and repair. The exposure was completed by inserting a Fukuda retractor to retract the humeral head. The anterior labrum and periosteum were excised, and the anterior glenoid surface was decorticated to create a flat bleeding surface of cancellous bone. The pegs of the suitable offset (4, 6, 8 mm) were engaged with the pre-drilled holes on the coracoid, allowing it to fit tightly against the overhanging offset bar for accurate graft positioning onto the glenoid.

The coracoid graft was ideally put between 3 and 5 o’clock on the glenoid, flushed to the articular surface and below the equator of the glenoid. The appropriate two 4 mm partially threaded cannulated screws were placed over the guide wires and tightened using a two-finger alternate technique. Over-tightening should be avoided to prevent a graft fracture. The guide wires were removed and the final graft position was rechecked. The capsule was repaired to the CAL stump with the arm in ABER (Figure 2).

### 2.4. Post-Operative Rehabilitation

All patients were immobilized by a cushioned 30° abduction brace to keep the limb in neutral rotation for 6 weeks. They were instructed to perform active exercises of the elbow, wrist, and fingers and passive ROM of the shoulder including external rotation starting at 30–40° of abduction at the first 3 weeks with a gradual increase to avoid external rotation loss. After the 6th week, they started to actively move the shoulder in all planes. Strengthening exercises, including biceps, triceps, internal and external rotators, and core stability training were delayed until 3 months post-operatively when the graft usually showed radiographic union. Heavy labor and sports were allowed at 6 months post-operatively. Regular visits were performed to instruct and follow up with the patients about ROM progress.

### 2.5. Post-Operative Evaluation

All patients were clinically assessed using the modified Rowe score for the evaluation of pain, stability, range of motion, and function of the shoulder [18]. A standard A/P X-ray was performed immediately, at 6 weeks, at 3 months post-operatively, and then at the end of follow-up. At 6 months follow-up, a CT 3D scan was performed to evaluate graft union and resorption (Table 3).

## 3. Results

A total of 27 patients with anterior shoulder instability and significant glenoid bone loss were initially enrolled. Six were lost to follow-up, resulting in a final cohort of twenty-one patients who underwent a modified Latarjet procedure and were included in the study. There were 18 males and 3 females with a mean age at the time of surgery of 28.52 ± 8.0 (range: 19–45) years. Twelve patients (57%) suffered recurrent instability in the dominant side and five patients (28.6%) were heavy workers. The mean number of dislocations before surgery was 18.33 ± 8.67 (range: 6–35) times. Seventeen patients (81%) had glenoid defects of 21 to 30%, while it was more than 30% in four patients (19%). The mean glenoid bone loss was 26.19 ± 4.85 (range: 21–37) %. All patients had HS lesions that were off-track in 19 cases (90.5%) and on-track in 2 cases (9.5%). Pre-operative radiological results are listed in Table 4 and Table 5.

### 3.1. Clinical Results

The mean post-operative follow-up period was 30.67 ± 7.53 (range: 16–40) months. Thirteen patients (61.9%) achieved excellent results and five patients (23.8%) showed good results, while two patients (9.5%) showed fair results and only one (4.8%) had a poor result. The mean modified Rowe score was increased from 60 ± 9.08 (range: 40–80) points pre-operatively to 85 ± 18.77 (range: 30–100) points post-operatively, and the difference was statistically significant (*p* = 0.001) (Figure 3).

Post-operatively, 18 patients (85.7%) had no pain, while 3 patients (14.3%) had mild to moderate pain. Sixteen patients had negative apprehension with no pain or subluxation, three had negative apprehension with pain in ABER, and two had positive apprehension with subluxation in ABER. No patients had a recurrent dislocation after surgery. Eleven patients (52.4%) had full ROM. The ROM was lost ˂25 % in any plane in nine patients and ˃25% in only one patient. Out of 19 patients who had no work limitation, 12 were able to practice sports at the same pre-injury level, 5 could practice sports but at a lower level, and 2 could no longer practice sports. Two patients had moderate work limitations. There were no cases of infection or neurological injuries (Table 6). 

The external rotation ranged between 40 and 60° with a mean of 48.33 ± 7.80° pre-operatively, while it ranged between 45 and 70° with a mean of 58.10 ± 7.50° post-operatively. The mean post-operative external rotation loss was 8.09 ± 5.11° (range: 0–20°). Post-operative forward elevation and external and internal rotation were significantly improved compared to pre-operatively (*p* = 0.001), while the difference with those of the contralateral normal shoulder was not statistically significant (Table 7).

### 3.2. Radiological Results

The coracoid graft was correctly positioned in sixteen patients (76.2%), whereas in five cases (23.8%), the coracoid position was wrong: in two patients it was too lateral, in the other two, it was too inferior, and in one case it was too medial. The screw insertion was also mal-positioned in two cases, but there was no screw breakage, loosening, and impingement (Figure 4). In three patients, the coracoid graft showed non-union and partial resorption at the end of follow-up.

Post-operative GH arthritis was not radiologically detected in 14 patients (66.7%), whereas it was a grade I in 5 patients (23.8%) and a grade II in 2 patients (9.5%), with no progression compared to pre-operative evaluation (Figure 5).

## 4. Discussion

The most important finding of this study is that the modified Latarjet procedure is a safe and effective technique in patients suffering from anterior shoulder instability with critical bone loss. This study showed that 85.7% of cases had satisfactory results with no recurrent dislocations and arthritis progression in a short-term follow-up. Majeed et al. [19] showed that 86% of cases had satisfactory results with a recurrence rate of 8%, whereas De Beer et al. [14] found that the recurrence rate was 4.9% after congruent-arc Latarjet. These excellent outcomes in terms of shoulder stability, despite significant defects in our patients, could be explained by the fact that the mean coracoid width is greater than its mean thickness. Therefore, the modified technique could reconstitute a significantly greater glenoid bone loss as compared with the classic orientation [20]. Some authors suggested that if glenoid defects exceed 30% of articular width or if the predicted glenoid track remains off-track with a classically performed Latarjet, a congruent-arc technique might be beneficial with its larger surface area [15,21]. In this study, all off-track HS lesions (19 cases) became on-track after a congruent-arc procedure. Instead, on the basis of theoretical calculations, if a classic Latarjet had been performed, only 13 off-track HS lesions would have become on-track. 

Moreover, the congruent-arc technique seems to show a lower risk of developing GH arthritis compared to a standard one. In fact, in a recent systematic review after standard Latarjet, patients showed a risk of 25.8 % of arthritis progression in a minimum of 5 years of follow-up [22]. On the contrary, in our study, 33.3 % of patients showed pre-operative radiological signs of GH arthritis, but with no progression at the final follow-up. This difference could be due to closely matched glenoid and coracoid surfaces resulting in decreased GH contact forces in the modified Latarjet, compared to the traditional one, despite a shorter follow-up of our study.

However, the precise etiology of GH arthritis in patients with traumatic anterior shoulder instability is still unknown. Many authors support the theory that considers the dislocation arthropathy to be a part of the natural history of shoulder instability [2]. Also in our study, the number of dislocations (more than 20 episodes), the time elapsed between the first dislocation and surgery (more than 5 years), and the older age at first dislocation significantly affected the incidence of GH arthritis.

Most authors reported a loss of external rotation between 9° and 12° after a Latarjet procedure, and in some cases, a loss of up to 20° [23]. In this study, the mean external rotation loss was 8.09 ± 5.11° (range: 0–20°). The external rotation loss might be caused by internal rotator contracture, coracoid malposition, tight capsular reinsertion, and adhesions in the anterior shoulder [24]. According to Allain et al. [25], the external rotation loss could be avoided when the subscapularis muscle splitting extends as medially as possible to avoid the impaction between the conjoined tendon and the muscle belly during external rotation, and maximizing the capsular length is also important for preserving the external rotation. In this study, using a subscapularis splitting approach instead of tenotomy, immediate post-operative rehabilitation including passive external rotation and repair of the capsule to the CAL stump with the arm positioned in external rotation could explain these good results.

All previous studies showed the importance of the graft position, which is directly related to the clinical results where a too-lateral position leads to arthritis and a too-medial one results in recurrent instability [26]. In our study, the coracoid graft was correctly positioned in 76.2 % of cases. This result is in contrast with the outcome of some other authors who reported a graft mal-positioning in more than 50% of cases [25,27]. This study reported better results than the previous literature due to the following reasons: firstly, the glenoid offset parallel drill guide was used for the easy control and positioning of the coracoid onto the glenoid; and secondly, the final position of the coracoid was checked by visualization and palpation, and if overhanged, the position was changed. The screw insertion was mal-positioned in two cases mainly due to the lower positioning of the graft and a lower learning curve in the first cases but there was no screw breakage, loosening, and impingement.

Matthes et al. [28] and Banas et al. [24] reported graft union in 76% and 82% of the patients, respectively. De Beer et al. [14] noted that partial graft resorption occurred in 9% of patients. The latter results were nearly in agreement with our study which showed that the graft union was noted in 18 patients (85.7%), whereas in 3 patients (14.3%), the coracoid graft showed non-union and partial resorption. These three patients showed good to excellent results, with only subjective apprehension felt by two of them. Therefore, graft non-union and resorption had no significant effect on the clinical outcome, which might be due to short-term follow-up and the limited number of cases that had non-union and resorption. The graft union could be enhanced by using a long piece of coracoid which is typically 2.5 to 3 cm, decorticating both the medial coracoid surface and glenoid rim and placing two bi-cortical compression screws parallel to the glenoid face [29].

However, the modified Latarjet technique showed some possible complications compared to the traditional one, although they did not occur in our study. The narrow coracoid margin is less tolerant of screw malposition and not suitable to safely accommodate two 4 mm screws, leading to a higher risk of weak fixation or intra-operative fracture [15]. Moreover, there is a greater risk of impingement in external rotation between the bone block and subscapularis tendon because it is larger than that in the classic procedure [30].

This study has several limitations. The small sample size limits the statistical power and generalizability of the results. Moreover, the absence of a control group prevents direct comparisons with alternative surgical techniques or conservative management. These aspects are inherent to the case series design. In addition, the short-term follow-up did not allow for a full assessment of long-term outcomes such as recurrence, glenohumeral arthritis progression, or graft resorption. 

## 5. Conclusions

Based on our short-term follow-up, the modified Latarjet procedure appears to be a safe, effective, and reliable surgical option for managing traumatic anterior shoulder instability with significant glenoid bone loss. It provided satisfactory results with no recurrent instability and arthritis progression. Most of the reported complications like external rotation loss, graft mal-positioning, and non-union associated with this procedure can be avoided with proper patient selection, systematic surgical technique, and adherence to an early rehabilitation protocol. This technique seems to reconstitute a significantly greater glenoid bone loss than the classic Latarjet, allowing greater contained translation of the humeral head. Additionally, the radii of the curvatures of the glenoid and coracoid inferior surfaces are closely matched, aiming to decrease the post-operative GH arthritis on a long-term follow-up, which needs to be further confirmed by other long-term studies, thus enhancing the functional outcome.

## Figures and Tables

**Figure 1 healthcare-13-01768-f001:**
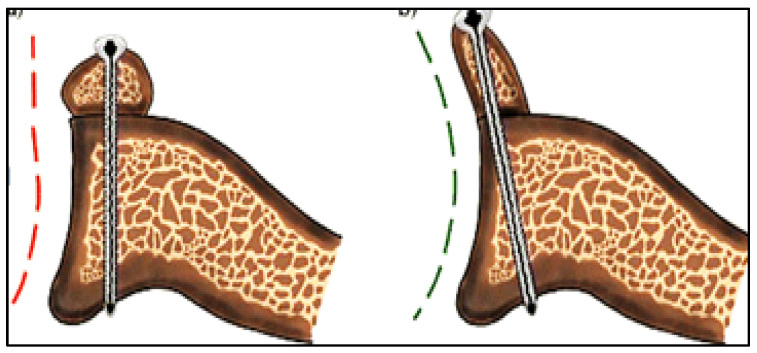
Traditional (red line) and congruent-arc Latarjet (green line).

**Figure 2 healthcare-13-01768-f002:**
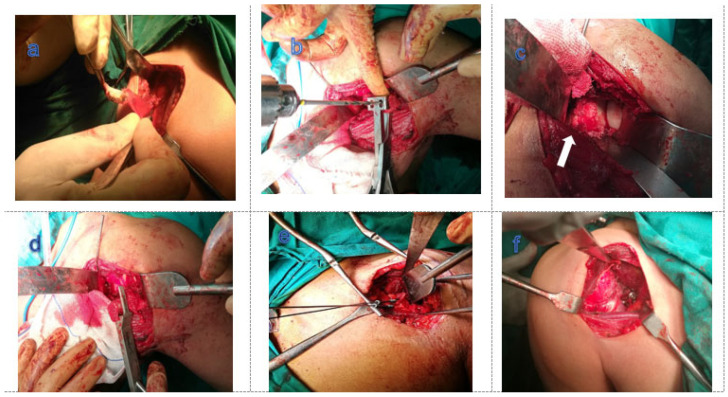
Surgical technique (**a**–**f**): (**a**) coracoid osteotomy with attached coracoacromial ligament; (**b**) grasping the coracoid-by-coracoid drill guide and drilling 2 parallel holes centered on the graft; (**c**) anterior glenoid rim preparation to obtain bleeding surface (white arrow); (**d**) engagement of coracoid with the glenoid offset; (**e**) placing and alternative tightening of 2 cannulated screws over the guide wires; (**f**) final coracoid fixation to the glenoid.

**Figure 3 healthcare-13-01768-f003:**
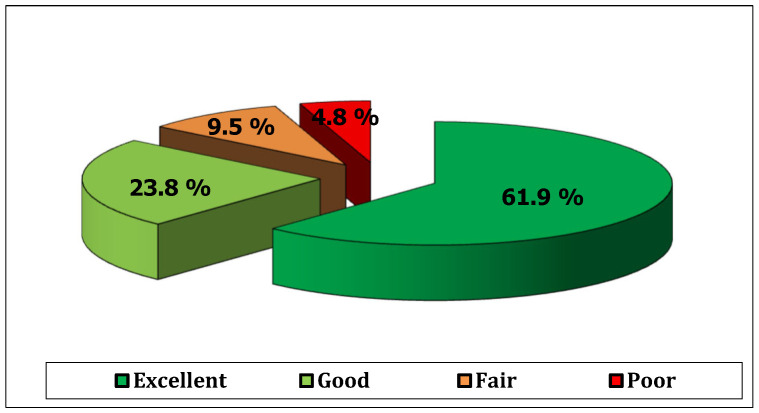
Post-operative clinical results.

**Figure 4 healthcare-13-01768-f004:**
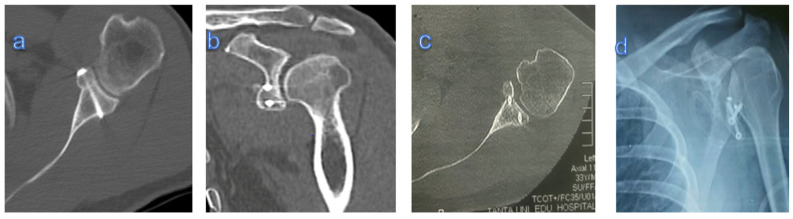
(**a**,**b**) Axial and coronal CT showing flushed united congruent graft; (**c**) axial CT showing too-lateral graft; (**d**) X-ray showing too-low graft and screw malposition.

**Figure 5 healthcare-13-01768-f005:**
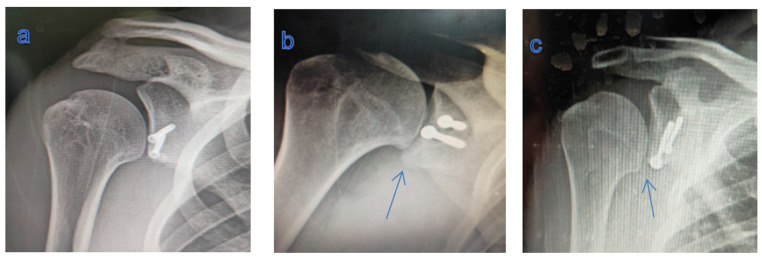
(**a**) X-ray showing no GH arthritis; (**b**,**c**) X-ray showing humeral osteophytes (blue arrows): grade I GH arthritis.

**Table 1 healthcare-13-01768-t001:** Inclusion and exclusion criteria.

Inclusion Criteria	Exclusion Criteria
Age > 18 years	First-time dislocation.
Chronic onset (>3 months)	Insignificant glenoid bone loss (˂21%).
Critical glenoid bone loss (˃21%) ± off-track HS lesion	Isolated Bankart lesion.
	Posterior, multidirectional, voluntary instabilities.
	Associated pathology as advanced GH arthritis, deltoid paresis, rotator cuff tear, and hyperlaxity.

HS: Hill–Sachs; GH: glenohumeral.

**Table 2 healthcare-13-01768-t002:** Pre-operative evaluation.

Category	Assessment	Purpose
**Clinical tests**	Apprehension and relocation tests	Confirm anterior instability
	Sulcus sign	Exclude multidirectional instability
	Beighton score	Screen for generalized hyperlaxity
**Functional**	Modified Rowe score	Baseline function, pain, stability, ROM
	Active range of motion (ROM)	Measure shoulder mobility
**Radiological**	A/P X-ray	Assess glenohumeral arthritis (Samilson–Prieto)
	3D CT scan	Quantify glenoid bone loss, Hill–Sachs lesion, and coracoid dimensions
	MRI	Evaluate labral, capsular, and rotator cuff pathology

AP: Anteroposterior, CT: computed tomography, and MRI: magnetic resonance imaging.

**Table 3 healthcare-13-01768-t003:** Post-operative evaluation.

Category	Assessment	Purpose
**Clinical**	Modified Rowe score	Assess outcome (pain, function, stability, ROM)
	Active range of motion (ROM)	Track recovery of mobility
**Radiological**	A/P X-ray	Evaluate graft position and arthritis progression
	3D CT scan (at 6 months)	Assess graft union and resorption

AP: Anteroposterior; CT: computed tomography.

**Table 4 healthcare-13-01768-t004:** Pre-operative radiological results: radiological parameters.

Parameter	Mean ± SD	Range
**Glenoid defect size (%)**	26.19 ± 4.85	21–37
**Coracoid width (mm)**	10.62 ± 1.60	9–14
**Coracoid thickness (mm)**	6.21 ± 1.63	4–10
**Glenoid coverage by coracoid width (%)**	41.33 ± 7.80	31–57
**Glenoid coverage by coracoid thickness (%)**	24.48 ± 7.28	16–40

**Table 5 healthcare-13-01768-t005:** Pre-operative radiological results: Hill–Sachs and glenohumeral arthritis.

Hill–Sachs Lesions		Glenohumeral Arthritis (Samilson–Prieto Classification)	
Type	n (%)	Grade	n (%)
**Off-track**	19 (90.5%)	**None**	14 (66.7%)
**On-track**	2 (9.5%)	**Grade I**	5 (23.8%)
		**Grade II**	2 (9.5%)

n: number.

**Table 6 healthcare-13-01768-t006:** Comparison of pre- and post-operative outcomes using modified Rowe score (n = 21).

Domain	Item	Pre-op.	Post-op.
**Pain**	No	15 (71.4%)	18 (85.7%)
	Mild to moderate	6 (28.6%)	3 (14.3%)
	Severe	0	0
**Stability**	No apprehension, no pain or subluxation	0	16 (76.2%)
	No apprehension, pain in ABER	0	3 (14.3%)
	Apprehension ± subluxation in ABER	21 (100%)	2 (9.5%)
**ROM**	Full ROM	9 (42.9%)	11 (52.4%)
	˂25% motion loss in any plane	10 (47.6%)	9 (42.9%)
	˃25% motion loss in any plane	2 (9.5%)	1 (4.7%)
**Function**	Return to pre-injury sport level	0	12 (57.1%)
	Return to pre-injury sport, not the same level	2 (9.5%)	5 (23.8%)
	Not return to pre-injury sport	11 (52.4%)	2 (9.5%)
	Moderate work limitations	6 (%)	2 (9.5%)
	Marked limitations (unable to work overhead)	2 (9.5%)	0
**Total**	Mean ± SD	60.0 ± 9.08	85.00 ± 18.77

ABER: abduction external rotation, ROM: range of motion, and SD: standard deviation.

**Table 7 healthcare-13-01768-t007:** Range of motion: pre-operative, post-operative, and normal contralateral side.

Movement	Pre-op	Post-op	Normal Shoulder
**Forward Elevation**	Range: 130–160°	Range: 150–175°	Range: 150–180°
	142.38 ± 8.31°	163.57 ± 7.44°	166.67 ± 8.56°
*p* value (Pre vs. Post)		0.001 *	
*p* value (Post vs. Normal)			0.222
**External Rotation**	Range: 40–60°	Range: 45–70°	Range: 50–80°
	48.33 ± 7.80°	58.10 ± 7.50°	66.19 ± 8.05°
*p* value (Pre vs. Post)		0.001 *	
*p* value (Post vs. Normal)			0.124
**Internal Rotation**	Range: 45–70°	Range: 60–80°	Range: 60–90°
	59.29 ± 8.41°	69.76 ± 7.15°	73.81 ± 9.21°
*p* value (Pre vs. Post)		0.001 *	
*p* value (Post vs. Normal)			0.119

*: Statistically significant.

## Data Availability

The datasets used and/or analyzed during the current study are available from the corresponding author on a reasonable request due to being stored on a secured internal network and actually not publicly available.

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
