# Peer review of "Congruent-Arc Latarjet Using Subscapularis Split Approach in the Treatment of Anterior Shoulder Instability with Significant Bone Loss: A Case Series"

_healthcare, 2025, doi:10.3390/healthcare13141768_

Round 1
Reviewer 1 Report
Comments and Suggestions for Authors
The manuscript addresses an important topic in orthopedic surgery, specifically the clinical outcomes of a modified Latarjet procedure in managing anterior shoulder instability with significant glenoid bone loss. The study is relevant and timely, given the ongoing debate on the indications and long-term results of congruent arc Latarjet techniques. However, several aspects of the manuscript require improvement to meet the standards of a high-impact medical journal.
Major Comments:
-
Study Design and Methodology:
-
The article is described as a “prospective” study, yet statistical methods are limited to descriptive analysis. Consider clarifying the study design (prospective observational case series?) and explicitly stating this in both the abstract and the methods.
-
The inclusion and exclusion criteria are reasonable but should be more clearly delineated in a dedicated paragraph or bulleted list for clarity.
-
-
Sample Size and Power:
-
The sample size of 21 patients is small. While acceptable for a preliminary case series, the authors should acknowledge this limitation more explicitly and ideally justify it with a sample size or power calculation if available.
-
-
Statistical Analysis:
-
The manuscript reports statistical significance (e.g., p = 0.001), but there is no description of the statistical tests used. This must be added to the methods section.
-
Include confidence intervals (CI) where appropriate, particularly for outcome scores.
-
-
Follow-up Duration:
-
The follow-up of 16–40 months is relatively short to assess post-operative arthritis progression. The authors should temper conclusions regarding arthritis prevention accordingly and suggest longer follow-up studies.
-
-
Radiographic Evaluation:
-
The classification used for assessing graft position (correct vs. malpositioned) is not defined. Provide a standardized scoring system or references for “correct” placement.
-
Similarly, the method for determining union vs. non-union and graft resorption should be clarified (CT criteria? Radiologist-reviewed?).
-
-
Discussion:
-
The discussion is generally well-structured but could benefit from clearer differentiation between findings from this study and those cited from literature.
-
Provide a more critical analysis of complications—e.g., although malpositioning occurred in 5 cases, it is unclear how this influenced outcomes.
-
-
Conclusion:
-
The conclusion overstates the results slightly. Consider rephrasing to reflect that the technique appears promising in the short term, pending further long-term data and larger cohorts.
-
Minor Comments:
-
Language and Style:
-
The manuscript would benefit from a professional English-language editing. Issues include inconsistent tense usage, occasional awkward phrasing, and punctuation errors.
-
Examples: “The lower bone defect threshold value for which isolated capsulo-labral repair is indicated remains controversial.” could be simplified and clarified.
-
-
Figures and Tables:
-
Figure legends should be more detailed (e.g., specify what the arrows point to, or what clinical relevance the images convey).
-
Table 1 and Table 2 need consistent formatting and more descriptive titles (e.g., “Table 2: Comparison of Preoperative and Postoperative Clinical Outcomes Using the Modified Rowe Score”).
-
-
References:
-
The references are generally appropriate, but some are cited incorrectly or redundantly (e.g., De Beer et al. [12] is mentioned multiple times—ensure consistency).
-
Confirm that all references follow the journal's formatting guidelines, including DOI links where possible.
-
-
Abbreviations:
-
The abbreviation list is helpful, but several terms (e.g., “GH,” “HS”) are used in the abstract before being defined—consider defining all abbreviations on first mention.
-
Recommendation:
Minor to moderate revision required before acceptance.
With careful attention to the above points—especially the methodological transparency, English editing, and improved data presentation—the manuscript could make a valuable contribution to the literature on shoulder instability surgery.
Author Response
Comments 1:
The manuscript addresses an important topic in orthopedic surgery, specifically the clinical outcomes of a modified Latarjet procedure in managing anterior shoulder instability with significant glenoid bone loss. The study is relevant and timely, given the ongoing debate on the indications and long-term results of congruent arc Latarjet techniques. However, several aspects of the manuscript require improvement to meet the standards of a high-impact medical journal.
Major Comments:
-
Study Design and Methodology:
-
The article is described as a “prospective” study, yet statistical methods are limited to descriptive analysis. Consider clarifying the study design (prospective observational case series?) and explicitly stating this in both the abstract and the methods.
-
The inclusion and exclusion criteria are reasonable but should be more clearly delineated in a dedicated paragraph or bulleted list for clarity.
-
-
Sample Size and Power:
-
The sample size of 21 patients is small. While acceptable for a preliminary case series, the authors should acknowledge this limitation more explicitly and ideally justify it with a sample size or power calculation if available.
-
-
Statistical Analysis:
-
The manuscript reports statistical significance (e.g., p = 0.001), but there is no description of the statistical tests used. This must be added to the methods section.
-
Include confidence intervals (CI) where appropriate, particularly for outcome scores.
-
-
Follow-up Duration:
-
The follow-up of 16–40 months is relatively short to assess post-operative arthritis progression. The authors should temper conclusions regarding arthritis prevention accordingly and suggest longer follow-up studies.
-
-
Radiographic Evaluation:
-
The classification used for assessing graft position (correct vs. malpositioned) is not defined. Provide a standardized scoring system or references for “correct” placement.
-
Similarly, the method for determining union vs. non-union and graft resorption should be clarified (CT criteria? Radiologist-reviewed?).
-
-
Discussion:
-
The discussion is generally well-structured but could benefit from clearer differentiation between findings from this study and those cited from literature.
-
Provide a more critical analysis of complications—e.g., although malpositioning occurred in 5 cases, it is unclear how this influenced outcomes.
-
-
Conclusion:
-
The conclusion overstates the results slightly. Consider rephrasing to reflect that the technique appears promising in the short term, pending further long-term data and larger cohorts.
-
Minor Comments:
-
Language and Style:
-
The manuscript would benefit from a professional English-language editing. Issues include inconsistent tense usage, occasional awkward phrasing, and punctuation errors.
-
Examples: “The lower bone defect threshold value for which isolated capsulo-labral repair is indicated remains controversial.” could be simplified and clarified.
-
-
Figures and Tables:
-
Figure legends should be more detailed (e.g., specify what the arrows point to, or what clinical relevance the images convey).
-
Table 1 and Table 2 need consistent formatting and more descriptive titles (e.g., “Table 2: Comparison of Preoperative and Postoperative Clinical Outcomes Using the Modified Rowe Score”).
-
-
References:
-
The references are generally appropriate, but some are cited incorrectly or redundantly (e.g., De Beer et al. [12] is mentioned multiple times—ensure consistency).
-
Confirm that all references follow the journal's formatting guidelines, including DOI links where possible.
-
-
Abbreviations:
-
The abbreviation list is helpful, but several terms (e.g., “GH,” “HS”) are used in the abstract before being defined—consider defining all abbreviations on first mention.
-
Recommendation:
Minor to moderate revision required before acceptance.
With careful attention to the above points—especially the methodological transparency, English editing, and improved data presentation—the manuscript could make a valuable contribution to the literature on shoulder instability surgery.
Response 1:
Thank you very much for your review and for the constructive comments regarding our manuscript. Your suggestions were extremely helpful in improving the quality of the study.
Major Comments
Study Design and Methodology:
We have added the study design as “prospective observational case series” both in the abstract and in the methods, as suggested. Additionally, we included a table listing the inclusion criteria to highlight them more clearly.
Sample Size:
We acknowledge the limited sample size and have emphasized this limitation in the discussion. Furthermore, we have added a statement underlining the need for larger studies and the inclusion of a control group in future research. We also added the number of patients lost to follow-up at the beginning of the Results section.
Statistical Analysis:
We added a paragraph at the end of the methods section specifying the statistical software used for data analysis. We also clarified the types of statistical calculations performed. The small sample size and the absence of a control group are inherent to the study design, which is a case series.
Follow-Up Duration
As stated in the Aim section, our study focused on a variety of outcomes, not only on the progression of glenohumeral osteoarthritis. However, we acknowledged in the limitations that the current follow-up duration is insufficient to assess the long-term progression of osteoarthritis. We added that further studies with longer follow-up periods and larger cohorts are planned. Additionally, our findings regarding the incidence of glenohumeral (GH) arthritis were intended as a short-term indicator suggesting that closely matched coracoid and glenoid surfaces may reduce arthritis incidence. We agree that this observation requires confirmation with longer follow-up.
Radiographic Evaluation:
The optimal graft position is described in the surgical technique section (flush and below the glenoid equator). Any deviation from this ideal position—either antero-posteriorly or supero-inferiorly (too lateral, too medial, overhanging, or too inferior)—was considered malposition. To our knowledge, no alternative classifications of graft malposition are available in the literature.
All findings were evaluated through standard radiographs and confirmed via 3D CT scan at 6 months. The diagnostic imaging protocol is clearly detailed in the postoperative evaluation section:
“A standard anteroposterior x-ray was performed immediately after surgery, at 6 weeks, at 3 months, and at the end of follow-up. A 3D CT scan was performed at the 6-month follow-up to evaluate graft union and resorption.”
Moreover, the radiological results section reports:
“The coracoid graft was correctly positioned in 16 patients (76.2%), whereas in 5 cases (23.8%), the position was incorrect: 2 were too lateral, 2 too inferior, and 1 too medial.”
Regarding graft resorption and non-union, only 3 patients were affected, which is a very small number. Non-union was clearly detectable on x-ray and confirmed via 3D CT. Partial graft resorption was identified based on decreased graft size and prominence of the screw head beyond the graft margin.
All radiological findings were confirmed by the same radiologist, whose evaluations were consistently documented.
Discussion:
Graft malposition occurred in 5 patients (23.8%) and had no impact on clinical outcomes for the following reasons:
- Our malposition rate compares favorably with rates reported in the literature (~50%), likely due to factors discussed in our manuscript.
- The medial and lateral graft deviations were minimal (a few millimeters), and thus insufficient to cause instability or GH arthritis.
- The suturing of the coracoacromial ligament to the capsule and the wider articular arc (coracoid being wider rather than thicker) likely contributed to stability and mitigated the effects of minor graft malposition.
Graft non-union and resorption occurred in 3 patients, all of whom had good to excellent clinical outcomes, with only 2 reporting subjective apprehension.
No cases of GH arthritis were observed postoperatively. The 7 cases of preoperative arthritis did not affect postoperative function.
In summary, the complications observed did not compromise clinical outcomes, likely due to the above-mentioned factors as well as the limited number of affected cases and relatively short follow-up period.
Conclusions:
The conclusion section has been revised according to your valuable recommendations.
Minor Comments
We have improved the clarity and completeness of the tables and figures. References have been reformatted to meet the journal’s style requirements. Abbreviations were removed from the abstract and added to the tables where appropriate.
Reviewer 2 Report
Comments and Suggestions for Authors
Dear authors, I am pleased to have the opportunity to review your manuscript on the impact of modified Latarjet on glenohumeral arthritis.
I would like to kindly comment on a few points:
1. Title: If the Latarjet modification is "congruent-arc," I recommend specifying this in the title or identifying the name of the technical variant used. If it is a new modification proposed by the authors, also include the evaluation of the effect on glenohumeral arthritis.
2. Abstract: I find it well expressed and structured.
3. Introduction: It is important to limit the scientific literature used to the last 5 years, which can reflect the current incidence of the pathology, not the one that existed in 2014, as in reference 1.
Likewise, there are concepts that must be clarified, such as values considered significant bone mass loss. The results of current studies that reflect the technique's results must be included, such as Efficacy and safety of the Latarjet procedure for the treatment of athletes with glenoid bone defects ≥ 20%: a single-arm meta-analysis https://doi.org/10.1186/s13018-024-04641-y, which compare it with other techniques and what the criteria for its use, benefits, and complications are. It is also important to introduce diagnostic and clinical criteria for glenohumeral arthritis and collect literature on the impact of surgery on the disease, including current international consensus reports, such as https://www.sciencedirect.com/science/article/abs/pii/S0749806321006964
4. Methods: They must specifically state that the study design is a case series, the clinical origin/setting of the participants and the surgery, who performed it, and the equipment. The clinical and surgical variables assessed prior to the intervention and to be compared after the intervention must be clear. Diagnostic tests are one thing to verify that the inclusion criteria are met; then the variables measured with both Rowe and both imaging tests must be clearly stated. Reference must be made to the literature to studies that have used these outcome measures with the disease studied and/or reference evidence of the validation and standardization of the instruments used.
The statistical methods used must be explained in accordance with the descriptive study.
4 Results: They should be expressed consistently in relation to the primary objective: assessing the outcome and progression of glenohumeral arthritis.
5 Discussion: I recommend reviewing the state of the art and comparing its results with more recent studies.
6 Conclusions: I find them well expressed, consistent with the data and objectives.
Author Response
Comments 2:
Dear authors, I am pleased to have the opportunity to review your manuscript on the impact of modified Latarjet on glenohumeral arthritis.
I would like to kindly comment on a few points:
1. Title: If the Latarjet modification is "congruent-arc," I recommend specifying this in the title or identifying the name of the technical variant used. If it is a new modification proposed by the authors, also include the evaluation of the effect on glenohumeral arthritis.
2. Abstract: I find it well expressed and structured.
3. Introduction: It is important to limit the scientific literature used to the last 5 years, which can reflect the current incidence of the pathology, not the one that existed in 2014, as in reference 1.
Likewise, there are concepts that must be clarified, such as values considered significant bone mass loss. The results of current studies that reflect the technique's results must be included, such as Efficacy and safety of the Latarjet procedure for the treatment of athletes with glenoid bone defects ≥ 20%: a single-arm meta-analysis https://doi.org/10.1186/s13018-024-04641-y, which compare it with other techniques and what the criteria for its use, benefits, and complications are. It is also important to introduce diagnostic and clinical criteria for glenohumeral arthritis and collect literature on the impact of surgery on the disease, including current international consensus reports, such as https://www.sciencedirect.com/science/article/abs/pii/S0749806321006964
4. Methods: They must specifically state that the study design is a case series, the clinical origin/setting of the participants and the surgery, who performed it, and the equipment. The clinical and surgical variables assessed prior to the intervention and to be compared after the intervention must be clear. Diagnostic tests are one thing to verify that the inclusion criteria are met; then the variables measured with both Rowe and both imaging tests must be clearly stated. Reference must be made to the literature to studies that have used these outcome measures with the disease studied and/or reference evidence of the validation and standardization of the instruments used.
The statistical methods used must be explained in accordance with the descriptive study.
4 Results: They should be expressed consistently in relation to the primary objective: assessing the outcome and progression of glenohumeral arthritis.
5 Discussion: I recommend reviewing the state of the art and comparing its results with more recent studies.
6 Conclusions: I find them well expressed, consistent with the data and objectives.
Response 2:
We would like to thank the reviewer for the valuable comments and suggestions, which have contributed significantly to the improvement of our manuscript.
In response to the feedback received, we have modified the title to better describe the surgical technique used. However, we did not include the effect on osteoarthritis in the title, as this was not the sole focus of our study.
We have updated the literature by including some more recent studies, and we have incorporated the two articles specifically suggested by the reviewer.
Additionally, we added two tables in the Methods section to more clearly distinguish between diagnostic criteria used for patient selection and the clinical and radiological outcome variables assessed before and after surgery. We clarified the role of the modified Rowe score and shoulder range of motion as validated outcome measures and specified the use of imaging studies both for diagnostic purposes (CT and MRI) and for the evaluation of surgical outcomes (graft position, union, and osteoarthritis progression). We also included a reference supporting the use of the Rowe score in the context of anterior shoulder instability.
Finally, we added a section detailing the statistical methods used, emphasized that the study was conducted as a case series, and provided additional details regarding the surgical technique, diagnostic assessments, and rehabilitation protocol.
Reviewer 3 Report
Comments and Suggestions for Authors
This prospective case series evaluates a modified Latarjet procedure (congruent arc technique with subscapularis split) for recurrent anterior shoulder instability
The abstract is structured; the introduction section should be shorter - including just an introductory phrase and the aim; state that no progression was observed, not just "None... showed... arthritis progression" (as pre-op arthritis existed); the keywords should be checked in accordance with MeSH.
In general, in the introduction section no figures should be provided. The mention of ISIS feels slightly disconnected. Briefly explain how ISIS relates to the decision for bony reconstruction in the context of this study's inclusion criteria. Please state the objective more clearly at the end of this section.
The methodology section is presented in subsection. A “Statistical analysis” subsection is missing as the statistical steps followed to obtain in the results is not clear. Propose a workflow algorithm with the modified Latarjet procedure taking example from Solyom A. et al. Clinical Workflow Algorithm. Provide more specifics on the passive ROM protocol (frequency, duration of sessions, progression criteria beyond "gradual increase"). Detail the strengthening program initiated at 3 months. "Soft tissue Bankart" is listed as exclusion. Does this exclude patients with combined significant bone loss and Bankart lesion? (This is the typical scenario).
In the results section please provide also some CT 3D scans with the follow-ups as mentioned in the methodology. The tables should include an abbreviation list at the end. Report p-values for all significant comparisons mentioned (e.g., pre/post ROM improvements - Table 3 mentions P=0.001 but doesn't specify for which motions).
The discussion section relates findings well to existing literature on stability outcomes, arthritis risk, ROM loss, and graft position/complications. The limitations paragraph should be extended (for example, the lack of a control group, emphasize that the small sample size etc. )
The conclusions are concise, but state that they are based on short-term follow-up.
The references are adequate but should be extended to reach a minimum of 30 given the type of paper.
Author Response
Comments 3:
This prospective case series evaluates a modified Latarjet procedure (congruent arc technique with subscapularis split) for recurrent anterior shoulder instability
The abstract is structured; the introduction section should be shorter - including just an introductory phrase and the aim; state that no progression was observed, not just "None... showed... arthritis progression" (as pre-op arthritis existed); the keywords should be checked in accordance with MeSH.
In general, in the introduction section no figures should be provided. The mention of ISIS feels slightly disconnected. Briefly explain how ISIS relates to the decision for bony reconstruction in the context of this study's inclusion criteria. Please state the objective more clearly at the end of this section.
The methodology section is presented in subsection. A “Statistical analysis” subsection is missing as the statistical steps followed to obtain in the results is not clear. Propose a workflow algorithm with the modified Latarjet procedure taking example from Solyom A. et al. Clinical Workflow Algorithm. Provide more specifics on the passive ROM protocol (frequency, duration of sessions, progression criteria beyond "gradual increase"). Detail the strengthening program initiated at 3 months. "Soft tissue Bankart" is listed as exclusion. Does this exclude patients with combined significant bone loss and Bankart lesion? (This is the typical scenario).
In the results section please provide also some CT 3D scans with the follow-ups as mentioned in the methodology. The tables should include an abbreviation list at the end. Report p-values for all significant comparisons mentioned (e.g., pre/post ROM improvements - Table 3 mentions P=0.001 but doesn't specify for which motions).
The discussion section relates findings well to existing literature on stability outcomes, arthritis risk, ROM loss, and graft position/complications. The limitations paragraph should be extended (for example, the lack of a control group, emphasize that the small sample size etc. )
The conclusions are concise, but state that they are based on short-term follow-up.
The references are adequate but should be extended to reach a minimum of 30 given the type of paper.
Response 3:
We thank the reviewer for the thorough evaluation and valuable suggestions, which helped us improve the quality and clarity of our manuscript.
We have revised the abstract by shortening the introductory section and highlighting the fact that no progression of glenohumeral arthritis was observed. Keywords have also been revised with greater attention to MeSH indexing.
The sentence regarding surgical indications has been rephrased as follows: “Balg and Boileau proposed that an Instability Severity Index Score (ISIS) > 6 is an indication for a bony procedure.”
The aim of our study is indeed the one described at the end of the introduction, as correctly pointed out.
We have added the missing section on statistical analysis and more clearly detailed the rehabilitation protocol at 3 months, as suggested. Regarding the workflow, since this is a case series, we believe it is premature to include one at this stage, and it would have limited scientific validity; however, we will consider including it in future studies with a larger sample size and longer follow-up.
Regarding soft tissue Bankart lesions, in our protocol, this represents an exclusion criterion only when it is the sole lesion present, we changed the term from "soft tissue Bankart lesion" to "isolated Bankart lesion" to avoid misunderstandings. However, if a soft tissue Bankart lesion is associated with significant glenoid bone loss, we do not consider it a true isolated soft tissue lesion, in accordance with recent literature.
We have revised and expanded the limitations section, clarified the conclusions to emphasize the short-term nature of our follow-up, and added additional references to strengthen the manuscript.
Round 2
Reviewer 2 Report
Comments and Suggestions for Authors
I would like to thank the authors for having carefully considered our comments and for providing thoughtful and appropriate responses. Their revisions have clearly strengthened the manuscript, which has improved notably as a result.
Reviewer 3 Report
Comments and Suggestions for Authors
The authors have improved their paper. Please edit the tables in APA academic style and change the article type.